# Protein-ligand binding representation learning from fine-grained interactions

**Shikun Feng**[1] **Minghao Li**[2] **Yinjun Jia**[3] **Weiying Ma**[1] **Yanyan Lan**[1,4†]
[1]Institute for AI Industry Research, Tsinghua University
[2]Beijing Institute of Genomics, Chinese Academy of Sciences
[3]School of Life Sciences, Tsinghua University
[4]Beijing Academy of Artificial Intelligence (BAAI)

## Abstract

The binding between proteins and ligands plays a crucial role in the realm of drug discovery. Previous deep learning approaches have shown promising results over traditional computationally intensive methods, but resulting in poor generalization due to limited supervised data. In this paper, we propose to learn protein-ligand binding representation in a self-supervised learning manner. Different from existing pre-training approaches which treat proteins and ligands individually, we emphasize to discern the intricate binding patterns from fine-grained interactions. Specifically, this self-supervised learning problem is formulated as a prediction of the conclusive binding complex structure given a pocket and ligand with a Transformer based interaction module, which naturally emulates the binding process. To ensure the representation of rich binding information, we introduce two pre-training tasks, i.e. atomic pairwise distance map prediction and mask ligand reconstruction, which comprehensively model the fine-grained interactions from both structure and feature space. Extensive experiments have demonstrated the superiority of our method across various binding tasks, including protein-ligand affinity prediction, virtual screening and protein-ligand docking.

## 1 Introduction

Understanding the interaction between proteins and ligands is a crucial task in drug discovery, which involves predicting whether the proteins and ligands can bind together or determining the binding affinity and pose of a protein-ligand pair. Deep learning methodologies (Öztürk et al., 2018; Abbasi et al., 2020; Monteiro et al., 2022; Wallach et al., 2015; Ragoza et al., 2017; Li et al., 2021b) have become prominent contenders for this direction due to recent and rapid advancements in machine learning. However, the performance of these data-driven methods heavily relies on limited training data and may be susceptible to noise introduced by experimental errors. Therefore, the overall generalizability of these supervised methods is constrained (Shen et al., 2021).

Inspired by the remarkable success of self-supervised learning in computer vision and natural language processing, recent works aim to apply it to protein-ligand interactions by utilizing large amount of unlabeled data. The majority of pre-training approaches available today, including ELECTRA-DTA (Wang et al., 2022a), SMT-DTA (Pei et al., 2022) and Uni-Mol (Zhou et al., 2023), rely on a two-tower architecture with individual molecular and protein encoders for pre-training. A simple interaction module is then introduced to fine-tune the encoders for downstream binding related tasks.

However, the binding mechanism of protein-ligand complex is exceedingly intricate, involving a broad range of non-covalent interactions between inter-molecular atom pairs, including $\pi$-stacking, $\pi$-cation, salt bridge, water bridge, hydrogen bond, hydrophobic interaction and halogen bond(de Freitas & Schapira, 2017). Previous studies (Adhav & Saikrishnan, 2023; Ding et al., 2022) have shown the crucial role of these interactions in determining binding affinity and docking conformation. Furthermore, certain supervised learning methods, for example MONN (Li et al., 2020), OnionNet (Zheng et al., 2019) and PIGNet (Moon et al., 2022), have demonstrated notable per-

---

[†]Correspondence to `lanyanyan@air.tsinghua.edu.cn`

formance improvement by explicitly incorporating these inter-molecular atom-wise interactions in their methodologies.

Clearly, the current self-supervised learning methods focus on enhancing the representation of individual molecule or protein, but the interaction module trained in the later fine-tuning stage falls short in capturing these highly intricate interaction patterns. Therefore, how to develop an interaction-aware representation that directly benefits downstream protein-ligand interaction-related tasks remains an unresolved challenge. To our best understanding, CoSP (Gao et al., 2022) represents a significant step forward in this direction by leveraging contrastive learning to obtain pocket and ligand representations. While the contrastive learning approach does have the capability to align positive ligand and pocket pairs, it fails to adequately capture the inter-molecular atomic interactions through the global matching manner.

To address this issue, we propose to learn protein-ligand binding representations from fine-grained interactions, named BindNet. Specifically, our self-supervised learning problem involves predicting the conclusive binding complex structure given a primary pocket and ligand, which is in line with the protein-ligand interaction process. To emphasize learning fundamental interactions, we employ a specific Transformer-based interaction module that utilizes individual pocket and ligand encoders in the modeling process. To ensure that the model learns interaction-aware protein and ligand representations, we use two distinct strategies in the pre-training process. The first pre-training objective is Atomic pairwise Distance map Prediction (ADP), where interaction distance map between atoms in the protein and ligand is employed to provide detailed supervision signals regarding their interactions. The other pre-training objective is Mask Ligand Reconstruction (MLR), in which the ligand representation extracted by a 3D pre-trained encoder is masked for reconstruction. By employing feature space masking and reconstruction instead of simply token or atom type masking, the model is more likely to capture richer semantic information, such as chemical and shape information, during the pre-training process, as has been demonstrated in prior works in the fields of computer vision and natural language processing (Baevski et al., 2023; Assran et al., 2023).

The primary contribution of this paper can be summarized in four distinct aspects. Firstly, the problem of self-supervised learning of protein-ligand binding representations has been formalized as the prediction of the final complex structure given the primary pocket and ligand structure, which naturally mimic the binding process. Secondly, a new architecture has been designed to incorporate a Transformer-based interaction module on protein and ligand encoders, emphasizing the encoding of intricate interaction representations rather than individual protein and ligand representations. Thirdly, two novel pre-training objectives have been proposed to ensure learning of complex binding patterns for the interaction module. Lastly, extensive experiments have been conducted on a wide range of binding-related tasks, including predicting protein-ligand affinity, virtual screening, and protein-ligand docking, all of which demonstrate the promising results achieved by BindNet.

## 2 RELATED WORK

Several pre-training methods have been proposed for proteins and ligands representation learning. DeepAffinity (Karimi et al., 2019) utilizes an RNN-based architecture to conduct unsupervised learning based on compound SMILES and protein SPS. Both SMT-DTA (Pei et al., 2022) and ELECTRA-DTA (Wang et al., 2022a) employ Masked Language Modeling (MLM) to train a molecule encoder based on SMILES and a protein encoder based on amino acid sequences. The distinction between these two methods is that SMT-DTA trains the MLM and the affinity prediction task concurrently in a multi-task fashion, whereas ELECTRA-DTA utilizes a pre-training and fine-tuning approach.

Aside from sequence-based methods, there are efforts to incorporate the 3D structure of ligands and pockets for modeling. Uni-Mol, for instance, employs denoising and MLM strategies to train a 3D-based molecular encoder and pocket encoder, which are then fine-tuned for prediction. While CoSP (Gao et al., 2022) leverages a contrastive learning framework on pocket-ligand pairs within unlabeled complex data, to pre-train a dual-branch encoder for pockets and ligands.

## 3 BINDNET

### 3.1 PROBLEM FORMALIZATION AND MODEL ARCHITECTURE

In order to capture the complicated protein-ligand interaction patterns, we have formulated the pre-training problem as directly predicting the final complex structure given the structures of both the

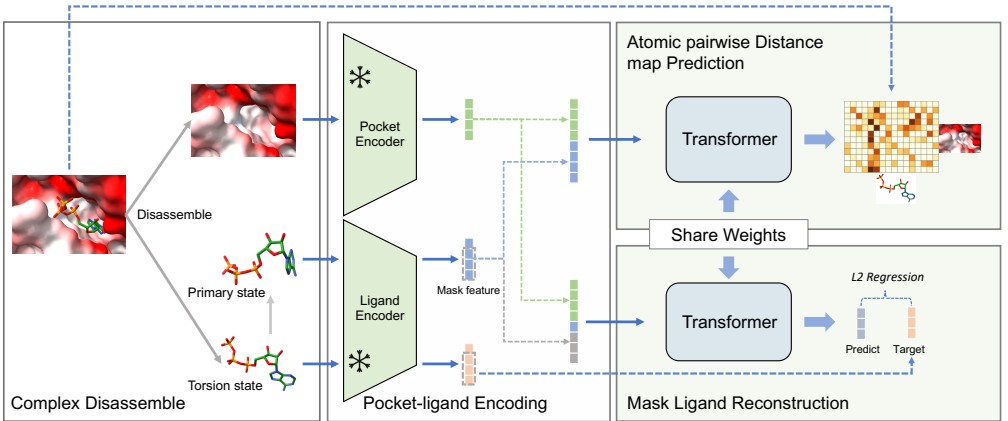

Figure 1: Illustrations of BindNet consisting of three components. **Left**: To generate an input for the model, the protein-ligand complex is first disassembled, yielding the individual pocket and ligand structures. Then the primary ligand structure is approximated via RDKit and perturbation techniques, and this structure is treated as one of the inputs for the model. **Middle**: Pocket and ligand encoders are employed to extract their embeddings, with the weights remaining frozen during both training and testing phases. **Right**: Two novel pre-training objectives, i.e. ADP and MLR, are introduced to learn binding representations.

pocket and ligand. In this formulation, the provided protein structure remains the same as that in the complex, while the given ligand structure represents its primary state, rather than its torsion state after binding. This formulation is more appropriate than using the torsion state as the input structure because it is commonly acknowledged that proteins tend to be relatively rigid, whereas molecules are generally more flexible in the binding process (Stärk et al., 2022; Corso et al., 2022).

Specifically, a structured chemical compound is denoted as $\mathcal{G} = (\mathcal{V}, \mathcal{X})$, where $v_i \in \mathcal{V}$ represents the atom type of node $i$, and $x_i \in \mathcal{X}$ represents the 3D position of the i-th atom. Then the full atom-level representation of a protein-ligand complex is denoted as $\mathcal{G}_C = (\mathcal{V}_C, \mathcal{X}_C)$. The pocket and ligand parts are denoted separately as $\mathcal{G}_P = (\mathcal{V}_P, \mathcal{X}_P)$ and $\mathcal{G}_L = (\mathcal{V}_L, \mathcal{X}_L)$, respectively. To ensure the primary molecular structure is used, RDKit is utilized to generate a stable conformation based on the ligand's chemical information. In case of any failures, we propose perturbing the torsion state to approximate the primary structure by introducing Gaussian noise to the dihedral angles or the coordinates of the molecular conformation. The resulted ligand structure is $\mathcal{G}_{\hat{L}} = (\mathcal{V}_{\hat{L}}, \mathcal{X}_{\hat{L}})$.

The architecture of BindNet is illustrated in Figure 1. Given a protein pocket and molecular ligand, two pre-trained encoders, designated as $\boldsymbol{\theta}_P$ and $\boldsymbol{\theta}_L$, are employed to secure preliminary representations for the pocket and ligand, denoted as $\mathbf{h}_P^{(0)}$ and $\mathbf{h}_L^{(0)}$, respectively. It is worth noting that the framework is versatile, and a variety of pre-existing encoders for pockets and ligands can be utilized. This paper uses the Uni-Mol encoders. It is important to understand that all pre-training and fine-tuning processes are conducted in the resulting representation space while keeping the two pre-trained encoders fixed, to emphasize the learning of the subsequent interaction module. More precisely, the interaction module, represented by $\boldsymbol{\theta}_I$, is an N-layer 3D-invariant Transformer that takes both atom-wise and pairwise representations as input and generates pocket and ligand representations $\mathbf{h}_P^{(N)}$ and $\mathbf{h}_L^{(N)}$, as well as pairwise binding representations $\mathbf{h}_{PL}^{(N)}$. These representations will be subsequently used in the fine-tuning process to perform various downstream tasks.

## 3.2 PRE-TRAINING OBJECTIVES

To facilitate the pre-training process and accurately capture the intricate binding process between proteins and ligands, two objectives have been proposed.

### 3.2.1 ATOMIC PAIRWISE DISTANCE MAP PREDICTION

According to previous biological and chemical studies (Ponder & Case, 2003; Alford et al., 2017), the energy that arises from various non-bond interactions between proteins and ligands is closely associated with their inter-molecular distances. Therefore, several score functions and deep learning

methods (Ballester & Mitchell, 2010; Zhu et al., 2020; Zheng et al., 2019; Moon et al., 2022) have utilized inter-molecular distance map to model the binding process, which has yielded significant improvements in binding related tasks.

Drawing inspiration from these findings, we propose utilizing the inter-molecular atom-wise distance map in a self-supervised manner to capture the intricate details of the interaction pattern. The inter-molecular distance matrix, denoted as $\mathcal{D}$, is initially derived from the original crystal structure of the complex. Each element, $d_{ij} \in \mathcal{D}$, represents the distance between the $i$-th atom in the pocket and the $j$-th atom in the ligand. Subsequently, we utilize the primary ligand $\mathcal{G}_{\hat{L}}$ and pocket $\mathcal{G}_P$ as inputs to predict the distance matrix $\mathcal{D}$ through regression. The specific objective function is:

$$\arg \min_{\boldsymbol{\theta}_I} \mathbb{E}_{(\mathcal{G}_P, \mathcal{G}_{\hat{L}})} \left[ \mathcal{L}_{reg} \left( \text{MLP} \left( \mathbf{h}_{P\hat{L}}^{(N)} \right), \mathcal{D} \right) \right], \tag{1}$$

where $\mathbf{h}_{P\hat{L}}^{(N)}$ represents the pair-wise embedding of $\mathcal{G}_P$ and $\mathcal{G}_{\hat{L}}$, $\mathcal{L}_{reg}$ denotes the $L_2$ regression loss.

Please note that our reason for using the primary ligand rather than the resultant ligand conformation in the complex data is to emphasize learning on the intricate interactions. Specifically, if we use the latter, this task focuses solely on learning the translation and rotation of the ligand to recover $\mathcal{D}$, thereby neglecting the crucial aspects of interaction information about inner changes in the ligand, such as variations in torsion angles when it binds to the target pocket.

### 3.2.2 MASK LIGAND RECONSTRUCTION

While the aforementioned ADP objective measures the binding of the original protein and ligand to form the final complex structure, the following MLR objective is designed to reflect the conditional dependency relations between protein and ligand representations in the binding process. More precisely, we randomly mask the representation of the torsion ligand state and reconstruct it from the representation of the entire pocket and the remaining atoms of the primary ligand.

To be specific, we replace the atom embeddings, denoted as $h_i$ in $\mathbf{h}_{\hat{L}}^{(0)}$ of the primary ligand with a learnable embedding $h_m$. Simultaneously, we mask the same corresponding set of atom embeddings in $\mathbf{h}_L^{(0)}$ of the torsion state ligand. These operations result in masked atom embeddings of torsion state ligand, as denoted as $\mathbf{h}_{L\mathbf{m}}^{(0)}$, which serves as the reconstruction target, along with the surrounding unmasked embeddings from the primary state ligand, denoted as $\mathbf{h}_{\hat{L}\backslash\mathbf{m}}^{(0)}$. Consequently, the objective of masked ligand reconstruction can be expressed as follows:

$$\arg \max_{\boldsymbol{\theta}_I} \mathbb{E}_{(\mathcal{G}_P, \mathcal{G}_{\hat{L}})} \left[ \text{P} \left( \mathbf{h}_{L\mathbf{m}}^{(0)} | \mathbf{h}_{\hat{L}\backslash\mathbf{m}}^{(0)}, \mathbf{h}_P^{(0)} \right) \right] \simeq \arg \min_{\boldsymbol{\theta}_I} \mathbb{E}_{(\mathcal{G}_P, \mathcal{G}_{\hat{L}})} \left[ \mathcal{L}_{reg} \left( \mathbf{h}_{L\mathbf{m}}^{(0)}, \tilde{\mathbf{h}}_{L\mathbf{m}}^{(0)} \right) \right], \tag{2}$$

where we employ $L_2$ regression loss for reconstructing the target embeddings, $\tilde{\mathbf{h}}_{L\mathbf{m}}^{(0)}$ denotes the corresponding atom embeddings predicted by $\boldsymbol{\theta}_I$.

Our masking strategy differs from traditional approaches in two ways. Firstly, the proposed mask and reconstruction methodology is carried out in the representation space, rather than at the atom token level in most previous work (Hu et al., 2019; Zhang et al., 2021; Li et al., 2021a). This is because atom types are usually limited in molecular applications, which makes the task trivial and the model easy to fit as demonstrated in Mole-BERT (Xia et al., 2022). While conducting mask and reconstruction in the representation space helps to capture the intricate interaction patterns between pocket and ligand. These patterns are not solely dependent on ligand atom types but also involve atom positions and contextual information, which have already been well captured in pre-trained molecular representations. Secondly, the remaining primary ligand representation is used to recover the masked torsion molecular representation, which captures both the conditional dependencies between protein and molecular features and the change in conformation during the binding process.

## 4 EXPERIMENTS

BindNet is pre-trained on BioLip (Yang et al., 2012), where we solely use the entries for regular ligands. For each complex, we extract the pocket-ligand segment by selecting residues within 8Å

distance from the ligand as the pocket. The complex is removed from our pre-training dataset if no residues are present within 8Å distance from the ligand or if only hydrogen atoms are present in the ligand. Finally, we obtain a dataset with 458,252 pocket-ligand complexes. The model optimization is carried out using the Adam optimizer with a learning rate of 1e-4 and a weight decay of 1e-4. The mask ratio of MLR is set to 0.8, and ADP and MLR losses are treated equally. The model is trained for 30 epochs with a batch size of 32 batch size, which is completed on a machine equipped with 8-A100 GPUs. This section demonstrates our experiments on various binding related downstream tasks, including protein-ligand binding affinity prediction, virtual screening, and molecular docking.

## 4.1 PROTEIN-LIGAND BINDING AFFINITY PREDICTION

Protein-ligand binding affinity prediction seeks to anticipate the degree of interaction strength between proteins and ligands. We assess the performance of BindNet on two binding affinity prediction related tasks, LBA and LEP, as originally proposed in Atom3D (Townshend et al., 2020).

### 4.1.1 LIGAND BINDING AFFINITY

**Data.** Ligand Binding Affinity (LBA) is a regression task that involves predicting the binding affinity value. The protein-ligand complexes and their associated binding strengths are obtained from the PDBBind dataset (Wang et al., 2005). The dataset is partitioned using a protein sequence identity threshold, resulting in two distinct splits: LBA 30% (with a protein sequence identity threshold of 30%) and LBA 60% (with a protein sequence identity threshold of 60%). We employ RMSE (Root Mean Square Error), Pearson correlation coefficient, and Spearman correlation coefficient, to evaluate BindNet. To ensure robustness of evaluation, we conduct three runs with different random seeds and report the mean values for the aforementioned metrics.

**Baselines.** We have compared BindNet with a diverse range of supervised methods, including sequence-based techniques such as DeepDTA, TAPE (Rao et al., 2019), and ProtTrans (Elnaggar et al., 2021); structure-based techniques, such as various variants of Atom3D, Holoprot (Somnath et al., 2021), and ProNet (Wang et al., 2022b); as well as semi/self-supervised methods namely DeepAffinity, SMT-DTA, GeoSSL (Liu et al., 2022) and Uni-Mol.

**Fine-Tuning BindNet.** Since the LBA dataset has provided precise crystal structural information on the binding complex, we utilize the binding pocket and ligand from this complex structure as input to conduct the fine-tuning process. Specifically, we select the embeddings of the CLS tokens from both pocket and ligand, which correspond to the first element of $\mathbf{h}_P^{(N)}$ and $\mathbf{h}_L^{(N)}$, respectively. These embeddings are concatenated and passed through a two-layer MLP to predict the binding affinity. The training process is supervised using a $L_2$ regression loss. Finally, we report the testing results based on the model that yields the best validation performance on the validation set.

**Results.** The experimental results are demonstrated in Table 1. Comparing different deep learning approaches, we observe that structure-based methods generally outperform sequence-based methods. This finding is rooted in the rich interaction information intrinsic to structural data, which offers more detailed insights than sequences. Furthermore, comparing pre-training methods to deep learning methods, recent advancements in pre-training, such as GeoSSL and Uni-Mol, can significantly outperform deep learning methods. This demonstrates the effectiveness of self-supervised learning using large amounts of unlabeled data. Importantly, our proposed BindNet achieves the best results in terms of all metrics for both LBA 30% and LBA 60%, indicating the benefit of capturing knowledge from fine-grained interactions, as compared to learning individual protein or ligand representations. Notably, BindNet performs particularly well in LBA 30%, which features strict data splitting and lower sequence identity between training and testing data. The substantial improvement underscores its superior generalizability by capturing essential interaction knowledge.

### 4.1.2 LIGAND EFFICACY PREDICTION

**Data.** Ligand Efficacy Prediction (LEP) is a task to classify whether a ligand activates a specific protein when provided with both the active and inactive structural states. We follow the split defined in Atom3D based on the protein function. Typical classification measures such as the Area Under

Table 1: Performance comparison of various methods on LBA dataset under different protein sequence identity split settings. The best and second-best results are highlighted in **bold** and underlined, respectively (all tables below are presented in this format).

| Methods | Model | LBA 30% | | | LBA 60% | | |
|---|---|---|---|---|---|---|---|
| | | RMSE↓ | Pearson↑ | Spearman↑ | RMSE↓ | Pearson↑ | Spearman↑ |
| Sequence based DL | DeepDTA | 1.866 | 0.472 | 0.471 | 1.762 | 0.666 | 0.663 |
| | TAPE | 1.890 | 0.338 | 0.286 | 1.633 | 0.568 | 0.571 |
| | ProtTrans | 1.544 | 0.438 | 0.434 | 1.641 | 0.595 | 0.588 |
| Structure based DL | Atom3D-CNN | 1.416 | 0.550 | 0.553 | 1.621 | 0.608 | 0.615 |
| | Atom3D-ENN | 1.568 | 0.389 | 0.408 | 1.620 | 0.623 | 0.633 |
| | Atom3D-GNN | 1.601 | 0.545 | 0.533 | 1.408 | 0.743 | 0.743 |
| | Holoprot | 1.464 | 0.509 | 0.500 | 1.365 | 0.749 | 0.742 |
| | ProNet | 1.463 | 0.551 | 0.551 | 1.343 | 0.765 | 0.761 |
| Pre-training Methods | DeepAffinity | 1.893 | 0.415 | 0.426 | - | - | - |
| | SMT-DTA | 1.574 | 0.458 | 0.447 | 1.347 | 0.758 | 0.754 |
| | GeoSSL | 1.451 | 0.577 | 0.572 | - | - | - |
| | Uni-Mol | 1.434 | 0.565 | 0.540 | 1.357 | 0.753 | 0.750 |
| | BindNet | **1.340** | **0.632** | **0.620** | **1.230** | **0.793** | **0.788** |

the Receiver Operating Characteristic (AUROC) and the Area Under the Precision-Recall Curve (AUPRC), are utilized as the evaluation metrics. Similar to LBA, we conduct three separate runs with varying random seeds and report the average results of the aforementioned metrics.

**Baseline Methods.** Our baseline methods include supervised techniques such as sequence-based approach DeepDTA and structure-based methods such as Atom3D-CNN, Atom3D-ENN, and Atom3D-GNN, along with pre-trained methods such as Uni-Mol and GeoSSL.

**Fine-Tuning BindNet.** We employ the pocket and ligand information provided by LEP as input, similar to LBA. Namely, we concatenate four pre-trained embeddings, incorporating the CLS tokens from the active structure's pocket and ligand, along with those of the inactive structure's pocket and ligand. Subsequently, these merged embeddings undergo a two-layer MLP for the final classification phase. The training process is supervised using cross-entropy. Finally, we report the testing results based on the model that yields the best validation performance on the validation set.

**Results.** Table 2 presents our experimental results. BindNet exhibits superior performance compared to all supervised learning and pre-training methods, as measured by both AUROC and AUPRC metrics. Notably, the improvement over the second-ranking method (Uni-Mol) is substantial, with an AUROC improvement of 0.882 vs. 0.823 and an AUPRC improvement of 0.870 vs. 0.787. These significant deviations further validate the effectiveness of focusing on learning binding representations, rather than individual protein and ligand representations.

Table 2: Comparison results on LEP datasets.

| Methods | Model | AUROC↑ | AUPRC↑ |
|---|---|---|---|
| Sequence based DL | DeepDTA | 0.696 | - |
| Structure based DL | Atom3D-CNN | 0.589 | 0.483 |
| | Atom3D-ENN | 0.663 | 0.551 |
| | Atom3D-GNN | 0.681 | 0.598 |
| | GVP-GNN | 0.628 | - |
| Pre-training Methods | GeoSSL | 0.776 | 0.694 |
| | Uni-Mol | 0.823 | 0.787 |
| | BindNet | **0.882** | **0.870** |

## 4.2 VIRTUAL SCREENING

**Data.** DUD-E (Mysinger et al., 2012) is a widely used benchmark for virtual screening, comprising 102 targets across multiple protein families. Each target contains an average of 224 active compounds and over 10,000 decoy compounds. We employ a 3-fold cross-validation for training

and evaluation, and our dataset split setting is consistent with AttentionDTI (Yazdani-Jahromi et al., 2022) and DrugVQA (Zheng et al., 2020), ensuring that similar targets are kept within the same fold to facilitate a fair comparison. Several widely used measures on DUD-E are employed in our evaluation, including AUROC and the ROC Enrichment metric (denoted as RE).

**Baseline Methods.** Various baseline methods are utilized in our experiments, including docking programs AutoDock Vina (Trott & Olson, 2010) and Smina (Koes et al., 2013), traditional statistical machine learning methods such as RF-score and NNScore (Durrant & McCammon, 2011), deep learning methods such as 3D-CNN, Graph-CNN (Torng & Altman, 2019), AttentionSiteDTI, DrugVQA, as well as pre-training methods such as Uni-Mol and CoSP.

Table 3: Performance comparison of different methods on DUD-E.

| Methods | Model | AUC↑ | 0.5% RE↑ | 1.0% RE↑ | 2.0% RE↑ | 5.0% RE↑ |
|---|---|---|---|---|---|---|
| Docking | AutoDock Vina | 0.716 | 9.139 | 7.321 | 5.811 | 4.444 |
| based | Smina | 0.696 | - | - | - | - |
| Scoring function | RF-score | 0.622 | 5.628 | 4.274 | 3.499 | 2.678 |
| based ML | NNScore | 0.584 | 4.166 | 2.980 | 2.460 | 1.891 |
| | 3D-CNN | 0.868 | 42.559 | 26.655 | 19.363 | 10.710 |
| Supervised | Graph-CNN | 0.886 | 44.406 | 29.748 | 19.408 | 10.735 |
| based DL | DrugVQA | **0.972** | 88.170 | 58.710 | 35.060 | **17.390** |
| | AttentionSiteDTI | 0.971 | 101.740 | 59.920 | 35.070 | 16.740 |
| Pre-training | CoSP | 0.901 | 51.048 | 35.978 | 23.681 | 12.212 |
| Methods | Uni-Mol | 0.945 | 82.586 | 50.206 | 30.162 | 14.789 |
| | BindNet | 0.960 | **105.277** | **61.602** | **35.150** | 16.185 |

**Fine-Tuning BindNet.** For each target, we extract residues within 6 Å distance from the crystal ligand as the pocket. We utilize cross-entropy based on the output of an MLP, with the CLS token embeddings of the pocket and ligand serving as inputs. Due to the significant imbalance between negative pairs (comprising inactive compounds) and positive pairs (comprising active compounds), we dynamically adjust the sampling weights to ensure that each batch contains an equal number of negative and positive samples. We report the mean performance of 3-fold cross-validation.

**Results.** As demonstrated in Table 3, BindNet achieves the highest performance with respect to three RE metrics and delivers competitive results in terms of the AUC score and the other RE metric. In particular, when compared to the other self-supervised learning methods such as Uni-Mol and CoSP, BindNet consistently outperforms them by a significant margin across all evaluation metrics. However, some supervised learning methods, such as DrugVQA and AttentionSiteDTI, display more stable and outstanding results. This phenomenon may be caused by the decoy bias hidden in the DUD-E dataset, as discussed in Gonczarek et al. (2016) and Chen et al. (2019). Specifically, the selection criteria for inactive compounds within DUD-E involve choosing compounds with similar physical properties to active compounds but differing topological structures, which may introduce a bias that emphasizes discrepancies between active and inactive compounds rather than focusing on protein-ligand interactions. Consequently, supervised learning methods may be more advantageous as they can directly incorporate and accommodate such biases.

To validate our assumption, we conduct further experiments on the AD dataset (Chen et al., 2019), which improves upon DUD-E by employing active compounds from other targets as decoys for the current target and effectively mitigates decoy bias. We evaluate the top four methods from the DUD-E dataset, and find a significant decrease in performance shown in Table 4, especially for supervised learning methods like DrugVQA and AttentionSiteDTI. This supports our claim that supervised methods tend to capture data biases rather than authentic interaction information. Conversely, Uni-Mol and BindNet perform notably better, with BindNet significantly surpassing Uni-Mol, reaffirming the criticality and benefit of learning intricate interaction patterns.

Table 4: Performance comparison on AD dataset.

| Methods | AUC↑ | 0.5% RE↑ | 1.0% RE↑ | 2.0% RE↑ | 5.0% RE↑ |
|---|---|---|---|---|---|
| DrugVQA | 0.48 | 3.00 | 2.44 | 2.15 | 1.79 |
| AttentionSiteDTI | 0.47 | 2.79 | 2.32 | 2.24 | 1.63 |
| Uni-Mol | 0.56 | 4.92 | 3.70 | 2.82 | 2.00 |
| BindNet | **0.64** | **6.98** | **4.45** | **3.21** | **2.97** |

### 4.3 MOLECULAR DOCKING

**Data.** We employ the same dataset as Uni-Mol to evaluate BindNet's performance in protein-ligand docking. The dataset includes the general set from PDBbind v2020 as the training set, with CASF-2016 as the test set, with data overlap removed for fair comparison. BindNet has learned the structural complex information in the pre-training stage, hence, we retrain it with the overlap removed data. Our evaluation metric involves calculating the Root Mean Square Deviation (RMSD) between predicted and actual positions, presenting the percentage of values falling below a threshold.

**Baselines.** Our baselines consist of a pre-training method Uni-Mol and various score-based methods, including AutoDock Vina, Vinardo (Quiroga & Villarreal, 2016), Smina, and AutoDock4 (Morris et al., 2009). These results are obtained directly from the original Uni-Mol paper.

**Fine-Tuning BindNet.** During the fine-tuning process, we initially predict the distance map of the docking complex, by using the pairwise representations $\mathbf{h}_{PL}^{(N)}$. Subsequently, we employ a post-processing approach to acquire the docked pose, by using gradient descent optimization, which is a common technique utilized in several typical docking methods (Lu et al., 2022).

Table 5: Performance comparison on docking pose prediction.

| Methods | 1.0 Å↑ | 1.5 Å↑ | 2.0 Å↑ | 3.0 Å↑ | 5.0 Å↑ |
|---|---|---|---|---|---|
| Autodock Vina | 44.21 | 57.54 | 64.56 | 73.68 | 84.56 |
| Vinardo | 41.75 | 57.54 | 62.81 | 69.82 | 76.84 |
| Smina | **47.37** | 59.65 | 65.26 | 74.39 | 82.11 |
| Autodock4 | 21.75 | 31.58 | 35.44 | 47.02 | 64.56 |
| Uni-Mol | 43.16 | 68.42 | **80.35** | 87.02 | 94.04 |
| BindNet | 45.26 | **69.82** | **80.35** | **89.12** | **94.38** |

**Results.** As depicted in Table 5, BindNet surpasses all other baseline methods. While Uni-Mol also outperforms all other score-based methods, it struggles to perform as effectively in the setting at the percentage under the 1.0Å threshold, compared to Smina and Autodock Vina. However, BindNet effectively manages to combat this issue by enhancing the percentage of results from Uni-Mol's 43.16% to BindNet's 45.26%. This indicates that BindNet's representations capture more precise interactions between ligands and proteins, leading to more accurate docking poses.

## 5 ABLATION STUDY

### 5.1 EFFECTS OF VARYING PRE-TRAINING OBJECTIVES

As BindNet incorporates two pre-training objectives, we conduct an ablation study to validate the impact of each loss using the LBA dataset, shown in Figure 2. It is evident that both ADP and MLR play crucial roles in BindNet's performance. The model trained solely on ADP or MLR outperforms the one without pre-training. Moreover, the best results are obtained by combining both strategies and training the model in a multi-task manner, demonstrating the complementary nature of these two strategies: ADP focuses on extracting binding knowledge from complexes, while MLR learns how to construct a ligand to form a stable binding pattern given a pocket target. The combination of these strategies results in a more comprehensive and robust interaction-aware representation.

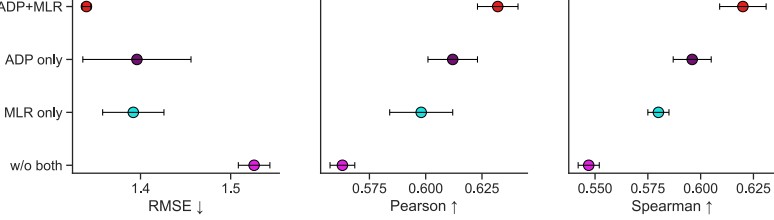

Figure 2: Ablation study on pre-training objectives, where the scatter represents the mean value of each metric and the error bar shows the standard deviation.

## 5.2 OUT-OF-DISTRIBUTION EVALUATION ON VARYING COMPLEX STRUCTURE

While 3D structure based deep learning methods often generate prediction results that are comparable to those of pre-training methods, they are usually evaluated in the in-distribution settings. In LBA, LEP and DUD-E, crystal structure, docking conformations and RDKit generated structures are used for both training and testing, respectively. As a result, it is unclear whether these deep learning models are capable of out-of-distribution generalization, a crucial issue in machine learning, particularly for robust and trustworthy learning in scientific fields.

We evaluate out-of-distribution capability with three new settings in LBA, with varying complex structure: CD (crystal conformations for training and docking complex conformations for testing), DD (docking complex conformations for training and testing), and RR (RDKit generated conformations for training and testing). The original setting CC uses crystal complex conformations for both training and testing. We utilize Atom3D-CNN and Atom3D-GNN as representative examples of 3D structure based supervised learning methods, exhibiting high performance on LBA (Table 1). As for pre-training methods, we evaluate Uni-Mol and BinNet. It should be noted that Atom3D-CNN and Atom3D-GNN are not evaluated in the RR setting as they only accept complex structure as input.

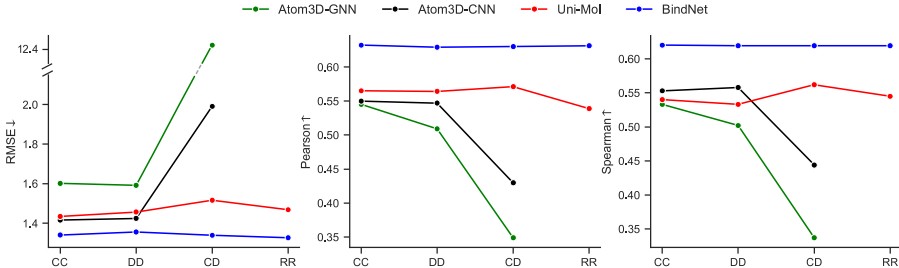

Figure 3: Performance of various methods across different settings with varying complex structure.

Figure 3 shows that pre-training methods exhibit superior out-of-domain generalization ability than supervised learning methods. Specifically, both Atom3D-CNN and Atom3D-GNN perform well under in-distribution settings CC and DD, but their out-of-distribution performance decreases significantly, as evidenced by the increase from 1.601 to 12.475 in RMSE when transitioning from CC to CD for Atom3D-GNN. Hence, one can infer that these 3D structure based supervised learning methods merely learn a data-fitting function, without truly capturing protein-ligand interaction patterns. Conversely, Uni-Mol and BindNet perform consistently well across all settings, even with randomly initialized conformations as input, demonstrating pre-training approach's efficacy in acquiring intrinsic data representations. Moreover, BindNet consistently outperforms Uni-Mol in both in-distribution and out-of-distribution evaluations, emphasizing the superiority of learning interaction-aware representations over individual protein and ligand representations.

## 6 CONCLUSION

This paper proposed a novel self-supervised pre-training method called BindNet for the purpose of learning protein-ligand binding representations. Unlike previous pre-training approaches that focus on individual protein and ligand representations, BindNet places greater emphasis on learning the binding representations using a Transformer-based interaction module, with fixed protein and ligand encoders as input. We proposed two new objective functions, i.e. ADP and MLR, to facilitate the pre-training from fine-grained interaction signals. Our analysis indicates that these objectives are crucial for learning comprehensive and robust interaction-aware representations, as they play complementary roles. By applying BindNet to various downstream binding related tasks, such as protein-ligand binding affinity prediction, virtual screening, and protein-ligand docking, we demonstrate that our approach significantly outperforms existing supervised and pre-training methods. Besides, our ablation study shows that BindNet successfully learns meaningful, robust representations that are capable of dealing with varying complex structures in out-of-distribution settings.

Although our primary focus is on the protein-ligand binding domain, the BindNet framework, as a powerful tool for learning binding representations, has great potential for extension to other bio-related binding tasks, such as protein-protein interactions and antigen-antibody recognition.

## ACKNOWLEDGEMENTS

This work is supported by the National Key R&D Program of China No.2021YFF1201600 and Beijing Academy of Artificial Intelligence (BAAI). We gratefully acknowledge Yanwen Huang for providing chemical knowledge consultation. Additionally, we extend our appreciation to the anonymous reviewers for constructive and helpful discussions.

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

## A    VISUALIZATION OF INTERACTION PATTERN LEARNED BY BINDNET

To elucidate the interaction pattern learned by BindNet, we select four protein-ligand complexes as illustrative examples to demonstrate the information captured by our interaction module in Figure 4.

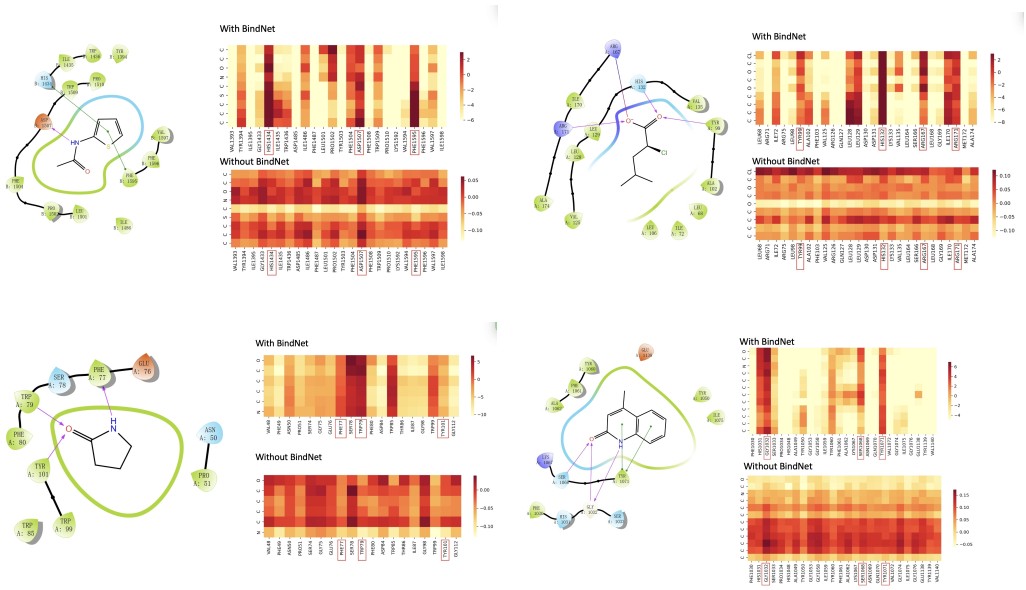

Figure 4: Visualization of the interaction patterns learned by BindNet.

The left part of each sub-figure illustrates interactions computed using Ligand Interaction Diagram Panel of Schrödinger (Schrödinger, LLC, 2023), depicting three common interactions: hydrogen interaction represented by the magenta line, $\pi$-$\pi$ interaction denoted by the green line, and salt bridge visualized as a gradient color line transitioning from red to blue.

The right part of each sub-figure represents the attention heatmap between pocket and ligand from BindNet. The original attention in BindNet is atom-atom level, which is too long to present here due to the significant number of atoms in the pocket, so we aggregate the attention of same residue in pocket to form an atom-residue level attention heatmap for a clearer visual representation. From these examples, we can easily find that BindNet pays more attention to the residue which can form non-covalent interactions with ligand, sometimes even the specific ligand atom-residue pair. This indicates BindNet might have the ability to capture the interaction force information between the pocket and ligand, which is crucial for tasks such as ligand binding affinity prediction and virtual screening.

## B    FLEXIBLE OF BINDNET

In this paper, we use molecular and pocket encoders pre-trained by Uni-Mol. However, our framework is flexible and allows for the integration of various pre-existing encoders for proteins and ligands. In this section, we make modifications, specifically by replacing both the molecular encoder and pocket encoder. This change aims to demonstrate the adaptability of our proposed framework to a variety of pre-existing encoders.

On the molecular side, we substitute the molecular encoder from Uni-Mol to incorporate the pre-trained encoders from Frad (Feng et al., 2023) and Coord (Zaidi et al., 2022), which are molecular pre-training models with a focus on quantum-related tasks. Notably, Frad and Coord's backbone is built on TorchMD-NET, a Graph Neural Network (GNN) that differs from Uni-Mol by not providing pair representations. The integration of Frad and Coord into our framework proved to be seamless,

as we can simply use the concatenation of corresponding node-level embeddings as pairwise representations. We reevaluate the performance of LBA. Furthermore, we present results for a scenario in which the interaction module remains unpretrained, as shown in Table 6.

On the protein side, we replace the pocket encoder with ESM2 (Lin et al., 2022) and ProFSA (Gao et al., 2023). In the case of ESM2, we adapt the protein input for the pocket structure to the protein sequence. Additionally, we provide results for the scenario in which the interaction module is not pre-trained based on ESM2 and ProFSA, as detailed in Table 6.

Table 6: Results for LBA task with different molecular and protein encoders with BindNet

| | LBA 30% | | | LBA 60% | | |
|---|---|---|---|---|---|---|
| Models | RMSE↓ | Pearson↑ | Spearman↑ | RMSE↓ | Pearson↑ | Spearman↑ |
| **Replace molecular encoder** | | | | | | |
| BindNet(Coord) wo pretrain | 1.447 | 0.545 | 0.533 | 1.515 | 0.653 | 0.656 |
| BindNet(Coord) with pretrain | 1.414 | 0.605 | 0.591 | 1.268 | 0.783 | 0.787 |
| BindNet(Frad) wo pretrain | 1.417 | 0.562 | 0.543 | 1.62 | 0.582 | 0.571 |
| BindNet(Frad) with pretrain | 1.386 | 0.612 | 0.597 | 1.341 | 0.761 | 0.762 |
| BindNet(Uni-Mol) wo pretrain | 1.434 | 0.565 | 0.541 | 1.357 | 0.753 | 0.753 |
| BindNet(Uni-Mol) with pretrain | 1.341 | 0.632 | 0.622 | 1.232 | 0.793 | 0.788 |
| **Replace protein encoder** | | | | | | |
| BindNet(ESM) wo pretrain | 1.561 | 0.444 | 0.445 | 1.521 | 0.664 | 0.682 |
| BindNet(ESM) with pretrain | 1.492 | 0.522 | 0.550 | 1.457 | 0.726 | 0.731 |
| BindNet(ProFSA) wo pretrain | 1.382 | 0.589 | 0.582 | 1.344 | 0.756 | 0.752 |
| BindNet(ProFSA) with pretrain | 1.359 | 0.611 | 0.596 | 1.286 | 0.786 | 0.786 |

Upon examining Table 6, it is evident that substituting the molecular or pocket encoder with Frad and Coord or ESM and ProFSA consistently results in superior performance for the pre-trained version of BindNet compared to the unpretrained version. Notably, BindNet, leveraging Frad and Coord as the molecular encoder and ProFSA as the protein encoder, consistently achieves SOTA results among other baselines, highlighting the inherent flexibility and effectiveness of our framework.

## C THE PERFORMANCE IMPACT RESULTING FROM THE OVERLAP BETWEEN PRE-TRAINING DATA AND DOWNSTREAM TASK DATA.

The BioLip data exhibits overlap with downstream tasks such as LBA and CASF. This overlap has the potential to influence performance and introduce a degree of bias in comparisons. However, the impact of this overlap is dependent on the objectives of the downstream tasks. In the case of the docking task, which aims to predict complex structures and is closely related to the pre-training target, it is imperative to eliminate the overlap to ensure a fair comparison. Conversely, for tasks involving binding affinity or virtual screening, where the goal is to predict binding affinity values or select positive protein-ligand pairs from a vast pool of negatives, the absence of such information as supervised signals means that the overlap may not significantly influence performance. To validate our hypothesis, we eliminate the overlap between the pre-training data and the two test datasets, which include LBA (30%) and the docking task. Subsequently, we conduct re-pre-training on both versions of the pre-training data and evaluate performance on the corresponding downstream tasks. The results are presented in the Table 7 and Table 8.

Table 7: Docking performance comparison between with and without overlap removal

| Models | 1.0 Å | 1.5 Å | 2.0 Å | 3.0 Å | 5.0 Å |
|---|---|---|---|---|---|
| BindNet | 46.68 | 72.98 | 80.35 | 90.87 | 95.79 |
| BindNet (remove overlap, reported in paper) | 45.26 | 69.82 | 80.35 | 89.12 | 94.38 |

As depicted in Table 7 and Table 8, the influence of overlap on the docking task performance is non-negligible, leading us to report the results in the second row (remove overlap version) in our paper. Conversely, for the binding affinity task, the disparity between the non-overlap and overlap is relatively marginal. Consequently, we opt to retain the overlap to align with other studies (Gao et al., 2022; Anonymous, 2023) that also utilized BioLip as pre-training dataset.

Table 8: Performance of Binding affinity prediction for LBA (30%) comparision with and without overlap removal

| Models | RMSE | Pearson | Spearman |
|---|---|---|---|
| BindNet | 1.340 | 0.632 | 0.620 |
| BindNet (remove overlap) | 1.396 | 0.625 | 0.616 |

# D LIGAND PRIMARY STATE GENERATION

We propose a data augmentation strategy employed during the training process, aiming to introduce random perturbations to the ligand structure. This serves not only to make the ADP task more challenging and meaningful but also to enhance the diversity of the pretraining data. Initially, we attempt to utilize RDKit to generate a random stable conformation based on the ligand's chemical information. In case of failure, we manually introduce rotations to the dihedral angles by applying Gaussian noise. Finally, if the ligand lacks rotatable bonds, we introduce Gaussian noise to the coordinates of the original conformer. The complete pseudocode for this algorithm is detailed in Algorithm 1.

---
**Algorithm 1** Data Augmentation of ligand conformation

---
**Require:**
$\quad \mathcal{G}_L = (\mathcal{V}_L, \mathcal{X}_L)$: Input ligand
$\quad \sigma$: Scale of dihedral angle noise
$\quad \tau$: Scale of coordinate noise
1: $X_{\hat{L}}$ = RDKitGenerate($\mathcal{G}_L$)   $\quad\quad\quad\quad\quad\quad\quad \triangleright$ Using RDKit to randomly generate a conformer
2: **if** $X_{\hat{L}}$ is not None **then**   $\quad\quad\quad\quad\quad\quad\quad\quad\quad\quad\quad\quad\quad\quad\quad\quad \triangleright$ Success
3: $\quad\quad$ return $\mathcal{G}_{\hat{L}} = (\mathcal{V}_L, \mathcal{X}_{\hat{L}})$
4: **else**
5: $\quad\quad$ **if** $\mathcal{G}_L$ has rotatable dihedral angles denoted as $\psi = (\psi_1, ..., \psi_m)^m$ **then**
6: $\quad\quad\quad$ Add Gaussian noise to each dihedral angles in $\psi$: $\psi_i = \psi + \Delta\psi, \Delta\psi \sim \mathcal{N}(0, \sigma^2)$ to get the new conformation $\mathcal{X}_{\hat{L}}$
7: $\quad\quad\quad$ return $\mathcal{G}_{\hat{L}} = (\mathcal{V}_L, \mathcal{X}_{\hat{L}})$
8: $\quad\quad$ **else**
9: $\quad\quad\quad$ $\mathcal{X}_{\hat{L}} = \mathcal{X}_L + \Delta\mathcal{X}$ , where $\Delta\mathcal{X} \sim \mathcal{N}(0, \tau^2 I_{3N})$, $N$ is atom number of $\mathcal{X}$
10: $\quad\quad\quad$ return $\mathcal{G}_{\hat{L}} = (\mathcal{V}_L, \mathcal{X}_{\hat{L}})$
11: $\quad\quad$ **end if**
12: **end if**

---

