# OpenReview forum: "Protein-ligand binding representation learning from fine-grained interactions"
_ICLR.cc/2024/Conference — ICLR 2024 poster_

### Official Review · Reviewer_7LCi · 2023-10-29

**Soundness:** 3 good
**Presentation:** 4 excellent
**Contribution:** 3 good
**Rating:** 6
**Confidence:** 5

**Summary:**

The paper introduces a novel approach named BindNet, designed for the pretraining of protein and ligand 3D representations. This method leverages two core objectives:

    Utilizing 3D representations of proteins and ligands sourced from Uni-Mol, BindNet aims to predict the pairwise atomic distances between atoms within a ligand-protein 3D complex.

    BindNet also employs protein 3D representations from Uni-Mol to predict masked ligand 3D representations, also obtained from Uni-Mol.

Both of these tasks are initially pretrained on a subset of the BioLiP dataset. Subsequently, the resulting protein and ligand representations are fine-tuned for tasks such as binding affinity prediction (LBA 30%, LBA 60%, LEP), virtual screening (DUDe, AD), and docking (CASF-2016). The performance of BindNet is compared against several baseline methods including UniMol, CoSP, GeoSSL, DeepAffinity, and SMT-DTA.

The study demonstrates that BindNet outperforms these baseline methods significantly in various downstream tasks. Additionally, the authors conduct an analysis to determine the importance of each pretraining objective in the process of representation learning.

While the experimental results are interesting, there are some concerns regarding the adequacy of leakage controls, a lack of comparison to stronger baselines, and the overall reproducibility of the research.

**Strengths:**

The experimental findings are interesting. It's the first time I've encountered a research work where the 3D protein-ligand complex is harnessed to learn a unified representation for both proteins and ligands, which has promising implications for practical binding affinity prediction.

Both of the proposed objectives for acquiring this joint representation yield compelling results, despite their simplicity. The paper is excellently written and presents a comprehensible narrative. The experiments encompass a wide range of tasks, including affinity prediction, docking, and virtual screening, and they feature thorough comparisons against numerous baseline techniques, including state-of-the-art methods in the field.

**Weaknesses:**

I hold a significant concern regarding the potential data leakage in the evaluation process. It appears that the representations learned from BioLiP, which contains a subset of PDBind used for validating the results, may introduce a form of leakage, thus giving pretrained methods an advantage over other baseline techniques. To address this, it is advisable to consider re-executing the experiments with the exclusion of all overlapping data between BioLiP and the downstream tasks, including overlaps with datasets for binding affinity prediction (LBA 30%, LBA 60%, LEP), virtual screening (DUDe, AD), and docking (CASF-2016).

Furthermore, I recommend including state-of-the-art protein representations like ESM-2 and ESM-1b (accessible at https://github.com/facebookresearch/esm), molecular representations such as MolFormer (available at https://github.com/IBM/molformer), and simple yet effective drug representations like Morgan fingerprints as baseline representations for a more comprehensive comparison.

Another significant concern is the reproducibility of the work. The absence of available code, missing details regarding hyperparameter settings, and the lack of transparency in releasing experimental settings, including dataset splits and configurations for downstream tasks, hinder the reproducibility of the research. Addressing these issues would greatly enhance the credibility and transparency of the work.

**Questions:**

1. Could you please kindly consider removing the potential leakage as advisable in the weakness discussion and release new results without leakage?
2. Could you please kindly include the comparison to state-of-the-art protein representations like ESM-2 and ESM-1b and molecular representations such as MolFormer and  Morgan fingerprints as baseline representations for a more comprehensive comparison?
3. Could you please kindly work-out on the reproducibility as advisable  the weakness discussion?

I would be very happy to consider revising my score when all the requests above are fulfilled.

---

> ### Author Response · Authors · 2023-11-20
>
> Thank you for your valuable suggestions. We acknowledge the need to conduct additional experiments to address the missing aspects you highlighted. We intend to incorporate this supplementary content in the revised version of our paper to provide a more comprehensive and complementary analysis. Interestingly, the questions you raised align with inquiries posed by other reviewers, further emphasizing the importance of addressing these points in the revised manuscript.
>
> >  Could you please kindly consider removing the potential leakage as advisable in the weakness discussion and release new results without leakage?(also asked by reviewer PEGf)
>
> Indeed, the Biolip data exhibits overlap with downstream tasks such as LBA and CASF. However, **the impact of this overlap is dependent on the objectives of the downstream tasks**. In the case of the docking task, which aims to predict complex structures and is closely related to the pretraining target, it is imperative to eliminate the overlap to ensure a fair comparison. Conversely, for tasks involving binding affinity or virtual screening, where the goal is to predict binding affinity values or select positive protein-ligand pairs from a vast pool of negatives, the absence of such information as supervised signals means that the overlap may not significantly influence performance.
> To validate our hypothesis, we eliminated the overlap between the pretraining data and the two test datasets, which include LBA (30%) and the docking task. Subsequently, we conducted re-pretraining on both versions of the pretraining data and reassessed performance on the corresponding downstream tasks. The results are presented in the tables below.
>
> To validate our hypothesis, we eliminated the overlap between the pretraining data and the two test datasets, which include LBA (30%) and the docking task. Subsequently, we conducted re-pretraining on both versions of the pretraining data and reassessed performance on the corresponding downstream tasks. The results are presented in the tables below.
>
> | Docking                  | 1.0 Å | 1.5 Å | 2.0 Å | 3.0 Å | 5.0 Å |
> | ------------------------ | ----- | ----- | ----- | ----- | ----- |
> | BindNet                  | 46.68 | 72.98 | 80.35 | 90.87 | 95.79 |
> | BindNet (remove overlap,the result reported in paper) | 45.26 | 69.82 | 80.35 | 89.12 | 94.38 |
>
>
> | Binding affinity prediction                  | RMSE  | Pearson | Spearman |
> | ------------------------ | ----- | ------- | -------- |
> | BindNet                  | 1.340 | 0.632   | 0.620    |
> | BindNet (remove overlap) | 1.396 | 0.625   | 0.616    |
>
> As depicted in the table above, the influence of overlap on the docking task performance is nonnegligible, leading us to report the results in the second row (remove overlap version) in our paper. Conversely, for the binding affinity task, the disparity between the non-overlap and overlap is relatively marginal, and this may be caused by our insufficient pre-training(only 6 epochs due to time limitation vs the orignial 30 epochs) . Consequently, we retained the overlap to maintain alignment with other studies [1], [2] that also utilized Biolip as a pretraining dataset.
>
> [1]CoSP: Co-supervised pretraining of pocket and ligand
>
> [2]General-purpose Pre-trained Model Towards Cross-domain Molecule Learning

---

> ### Author Response · Authors · 2023-11-20
>
> > Could you please kindly include the comparison to state-of-the-art protein representations like ESM-2 and ESM-1b and molecular representations such as MolFormer and Morgan fingerprints as baseline representations for a more comprehensive comparison?
> (asked by all reviewers)
>
> Thank you for your valuable suggestion. This particular question has been raised by all three reviewers. To answer this question, we have made significant modifications, specifically by replacing both the molecular encoder and pocket encoder. This change aims to demonstrate the adaptability of our proposed framework to a variety of pre-existing encoders.
>
> On the molecular side, we have substituted the **molecular encoder** from UniMol to the pretrained encoder provided by **Frad**[1], which is a molecular pretraining model with a focus on quantum-related tasks. Frad's backbone is built on torchmd-net, which is a gnn and does not output pair representations like 'unimol.' Integrating Frad into our framework is seamless as we can simply use the concatenation of corresponding node-level embeddings as pairwise representations. We conducted pretraining of the interaction module based on Frad for 8 epochs, considering time limitations. We reevaluated the performance of LBA, and additionally, we present results for the setting where the interaction module is unpretrained, as shown in the table below.
>
> On the protein side, we replaced the **pocket encoder** with the **ESM2**[2] encoder and altered the protein input for pocket structure to the protein sequence. Pretraining was also conducted based on the ESM2 encoder. Similarly, we include results for the setting where the interaction module is unpretrained based on the ESM2 encoder, as detailed in the table below.
> | idx | Methods                                | LBA(30%) RMSE | LBA(30%) Pearson | LBA(30%) Spearman | LBA(60%) RMSE | LBA(60%) Pearson | LBA(60%) Spearman |
> | --- | -------------------------------------- | ------------- | ----------------- | ------------------ | ------------- | ----------------- | ------------------ |
> |     | **Replace molecular encoder to Frad**  |               |                   |                    |               |                   |                    |
> | 0   | BindNet(Frad encoder) wo pretrain      | 1.417         | 0.562             | 0.543              | 1.62          | 0.582              | 0.571              |
> | 1   | BindNet(Frad encoder) with pretrain    | 1.386         | 0.612              | 0.597              | 1.341         | 0.761              | 0.762               |
> | 2   | BindNet(UniMol) wo pretrain            | 1.434         | 0.565             | 0.541               | 1.357         | 0.753             | 0.753               |
> | 3   | BindNet(UniMol encoder) with pretrain  | 1.341          | 0.632             | 0.622               | 1.232          | 0.793             | 0.788              |
> |     | **Replace pocket encoder**             |               |                   |                    |               |                   |                    |
> | 4   | BindNet(esm2 encoder) wo pretrain      | 1.561         | 0.444             | 0.445              |               |                   |                    |
> | 5   | BindNet(esm2 encoder) with pretrain    | 1.492         | 0.522             | 0.550               |               |                   |                    |
>
> Upon examining the table above, it is evident that replacing the molecular or pocket encoder with Frad or ESM consistently results in superior performance for the pretrained version of Bindnet compared to the unpretrained version (idx0 vs idx1, idx4 vs idx5). Notably, Bindnet based on Frad continues to achieve SOTA results among other baselines, highlighting the flexibility of our framework.
>
>
> It is essential to draw attention to the ESM2 encoder, which has nearly 3 billion parameters. Due to limitations in computational resources and time constraints, we were compelled to randomly select residues when the sequence number exceeded 500, and the model was trained for only 4 epochs.  Such constraints may have harmful effects on its overall performance. These constraints may have deleterious effects on its overall performance. We firmly believe that alleviating these constraints could significantly enhance its performance.
>
> [1] Fractional Denoising for 3D Molecular Pre-training. ICML 2023
>
> [2] Language models of protein sequences at the scale of evolution enable accurate structure prediction.

---

> ### Author Response · Authors · 2023-11-20
>
> > Could you please kindly work-out on the reproducibility as advisable the weakness discussion?
>
> We have reorganized our code and uploaded it to an anonymous Git repository. Additionally, we have provided a comprehensive README file that includes both the pretraining and fine-tuning  and data processing scripts, complete with the hyperparameters utilized during training. You can access the code repository through this link:https://anonymous.4open.science/r/BindNet-0060/README.md . Furthermore, we have included a pretrained model, available at the provided link(https://drive.google.com/file/d/1k40YI_Et472S3lG1CMs5-of4-Gf_op8_/view?usp=sharing), we also upload the processed data(https://drive.google.com/drive/folders/1NfXpsbbsR7FNIEoU88EA04Jmgnio0q_O?usp=sharing) . If you have any further questions regarding reproducibility or if there are specific details you would like us to expand upon, please feel free to ask. We welcome any inquiries and are eager to provide additional information as needed.

---

> ### Author Response · Authors · 2023-11-21
>
> Dear reviewer:
> We are deeply grateful for your invaluable suggestions aimed at enhancing our manuscript. We kindly seek your confirmation on whether any remaining issues require further attention to meet your expectations and potentially elevate the overall assessment. Your time and feedback are immensely valued, and we eagerly await your response.

---

> ### Comment · Reviewer_7LCi · 2023-11-22
> **Thank you for your response**
>
> Dear authors,
> Thank you for your responses and additional experimental results on removing potential leakages, that result is interesting, please update the results in the paper with leakage removed.
> Regarding the experiment with ESM-2, it is a large model, why didn't you try with ESM-1b which is a smaller one?
> Regarding the experiment with molecular encoder, it seems that molecular representation plays an important role so the baselines should include some recent molecular representation learning method as I suggested.
> Based on the new experimental results I  have raised my score.
> Best regards,

---

> > ### Author Response · Authors · 2023-11-22
> > **Thanks for your feedback**
> >
> > Dear reviewer,
> >
> > We extend our sincere gratitude for your invaluable suggestions, which have significantly enriched the content of our paper. We are committed to incorporating the discussed improvements into our revised manuscript.
> >
> > Regrettably, due to time constraints, we find ourselves unable to fulfill the experiment request to replace additional molecule encoders before the impending rebuttal deadline. **The re-pretraining process, which is essential for replacement, requires a couple of days to complete**. Nevertheless, we believe that the replacement of Frad and ESM experiments adequately justifies the flexibility of our framework, showcasing its compatibility with various pre-trained methods. **Our primary focus lies in designing  pre-training strategies for learning interaction-aware representations, which is beneficial for binding-related downstream tasks. The results presented thus far indicate that whether applying BindNet pretraining or not makes a difference to performance no matter which pocket or molecular encoder it uses. And we also supplement more visual analysis to show that BindNet has captured specific types of interactions**( Please check the answer to the first question of Reviewer kw5e: https://openreview.net/forum?id=AXbN2qMNiW&noteId=gddLSiB5lo) . Therefore, we contend that additional experiments involving the replacement of more molecular encoders, given the substantial pre-training costs involved, may not significantly contribute to the relevance or significance of our method within the limited time during rebuttal period.
> >
> >
> > Nevertheless, we assure you that these experiments remain on our agenda, and we want to take more effort to improve the ESM version of BindNet's performance (replace the esm encoder to a smaller version or use the offline embeddings ) and explore additional molecular encoders to expand the application scope of BindNet. Whether our paper is accepted or not, we look forward to supplementing it with more interesting results that demonstrate the continued evolution of our research.
> >
> > Thank you once again for your insightful feedback!
> >
> > Best regards.

---

> ### Author Response · Authors · 2023-11-23
>
> Dear reviewer:
> We have included additional test results for the ESM version of BindNet on the LBA (60%) task to enhance the completeness of our experiments, which was previously absent in the previous answers. Similar to the observations in the LBA (30%) task, the application of BindNet demonstrates an enhancement in performance when utilizing the ESM encoder. This improvement signifies the benefits of our proposed method.
>
> | idx | Methods                                | LBA(30%) RMSE | LBA(30%) Pearson | LBA(30%) Spearman | LBA(60%) RMSE | LBA(60%) Pearson | LBA(60%) Spearman |
> | --- | -------------------------------------- | ------------- | ----------------- | ------------------ | ------------- | ----------------- | ------------------ |
> |     | **Replace molecular encoder to Frad**  |               |                   |                    |               |                   |                    |
> | 0   | BindNet(Frad encoder) wo pretrain      | 1.417         | 0.562             | 0.543              | 1.62          | 0.582              | 0.571              |
> | 1   | BindNet(Frad encoder) with pretrain    | 1.386         | 0.612              | 0.597              | 1.341         | 0.761              | 0.762               |
> | 2   | BindNet(UniMol) wo pretrain            | 1.434         | 0.565             | 0.541               | 1.357         | 0.753             | 0.753               |
> | 3   | BindNet(UniMol encoder) with pretrain  | 1.341          | 0.632             | 0.622               | 1.232          | 0.793             | 0.788              |
> |     | **Replace pocket encoder**             |               |                   |                    |               |                   |                    |
> | 4   | BindNet(esm2 encoder) wo pretrain      | 1.561         | 0.444             | 0.445              |  1.521             |      0.664             |    0.682                |
> | 5   | BindNet(esm2 encoder) with pretrain    | 1.492         | 0.522             | 0.550               |        1.457       |         0.726          |     0.731               |

---

> ### Author Response · Authors · 2023-11-23
> **Revised version of paper**
>
> Thanks for your valuable time spent reviewing our paper. We have thoroughly addressed all of your questions in the following responses and clarified some aspects in the revised version of the paper:https://anonymous.4open.science/api/repo/BindNet_rebuttal-88C1/file/BindNet_revised.pdf. The primary updates can be found **in the appendix and highlighted in blue** for your reference.

---

### Official Review · Reviewer_PEGf · 2023-10-30

**Soundness:** 2 fair
**Presentation:** 1 poor
**Contribution:** 2 fair
**Rating:** 6
**Confidence:** 3

**Summary:**

This paper present a new training methodology with two novel training loss: the atomic pairwise distance prediction and mask ligand reconstruction. The trained model then fine-tuned on down-stream tasks like binding affinity prediction and virtual screening. The experimental results seem encouraging.

**Strengths:**

- Overall, the paper is clearly represented.
- The authors conducted multiple down-stream tasks to verify the efficacy of their method.

**Weaknesses:**

- The authors claim that it is a "self-supervised" method, but actually the pocket-ligand complex data are still required for pre-training. So, the data volume is limited by this requirement.  For example, only 458k BioLip data is used for pre-training, and thus earlier stage pretraining like UniMol is required.

- Another concern is about label leakage. Is there any overlap between the BioLip and downstream datasets (LBA, DUD-E, PDBBind, etc.)?

- The authors claim that "a variety of pre-existing encoders for pockets and ligands can be utilized. ", but only Uni-Mol used in experiments.

- Using RDKit to build the primary state is not accurate and sometimes even fail. The authors are encouraged to use more advanced methods for this.

**Questions:**

- What is the " N-layer 3D-invariant Transformer" used in the paper? The authors should provide more details about this part.
- How to choose the mask ratio in MLR? The 0.8 seems to be a very high value.

---

> ### Author Response · Authors · 2023-11-20
>
> Thank you for reviewing our paper. We appreciate your expert feedback, insightful concerns, and have carefully considered and explained your suggestions where necessary.
>
> >  The authors claim that it is a "self-supervised" method, but actually the pocket-ligand complex data are still required for pre-training. So, the data volume is limited by this requirement. For example, only 458k BioLip data is used for pre-training, and thus earlier stage pretraining like UniMol is required.
>
> The Biolip data is also taken as pre-training dataset for other self-supervised methods[1] [2]. While it is recognized that the volume of complex data is relatively limited compared to single molecule or protein data, we notice that a recent method called ProFSA[3] introduces a novel approach to generate simulated complex data utilizing the existing protein structures, which contains approximately **5 million** data items. To validate the adaptability of our method to such "pseudo" complex data, we conducted pretraining and assessed its performance on the LBA 30% task. The results are presented in the table below.
>
> | Methods                                      | RMSE  | Pearson | Spearman |
> | -------------------------------------------- | ----- | ------- | -------- |
> | GeoSSL                                       | 1.451 | 0.577   | 0.572    |
> | BindNet (pretrained on ProFSA data, 4 epochs) | **1.337** | 0.614   | 0.591    |
> | BindNet (pretrained on biolip, 30 epochs)     | 1.34  | **0.632**   | **0.62**     |
>
> As depicted in the table above, BindNet pretrained on the ProFSA data still achieves state-of-the-art (SOTA) results when compared to other baseline models. It is worth noting that the pretraining process was limited to just 4 epochs due to time constraints during the rebuttal phase and the big scale of pretraining data. Despite this limitation, the model has the potential to further improve in performance with more training iterations.
>
> [1] CoSP: Co-supervised pretraining of pocket and ligand
>
> [2] General-purpose Pre-trained Model Towards Cross-domain Molecule Learning
>
> [3] Self-supervised Pocket Pretraining via Protein Fragment-Surroundings Alignment
>
> > Another concern is about label leakage. Is there any overlap between the BioLip and downstream datasets
>
> Indeed, the Biolip data exhibits overlap with downstream tasks such as LBA and CASF. However, **the impact of this overlap is dependent on the objectives of the downstream tasks**. In the case of the docking task, which aims to predict complex structures and is closely related to the pretraining target, it is imperative to eliminate the overlap to ensure a fair comparison. Conversely, for tasks involving binding affinity or virtual screening, where the goal is to predict binding affinity values or select positive protein-ligand pairs from a vast pool of negatives, the absence of such information as supervised signals means that the overlap may not significantly influence performance.
> To validate our hypothesis, we eliminated the overlap between the pretraining data and the two test datasets, which include LBA (30%) and the docking task. Subsequently, we conducted re-pretraining on both versions of the pretraining data and reassessed performance on the corresponding downstream tasks. The results are presented in the tables below.
>
> To validate our hypothesis, we eliminated the overlap between the pretraining data and the two test datasets, which include LBA (30%) and the docking task. Subsequently, we conducted re-pretraining on both versions of the pretraining data and reassessed performance on the corresponding downstream tasks. The results are presented in the tables below.
>
> | Docking                  | 1.0 Å | 1.5 Å | 2.0 Å | 3.0 Å | 5.0 Å |
> | ------------------------ | ----- | ----- | ----- | ----- | ----- |
> | BindNet                  | 46.68 | 72.98 | 80.35 | 90.87 | 95.79 |
> | BindNet (remove overlap,the result reported in paper) | 45.26 | 69.82 | 80.35 | 89.12 | 94.38 |
>
>
> | Binding affinity prediction                  | RMSE  | Pearson | Spearman |
> | ------------------------ | ----- | ------- | -------- |
> | BindNet                  | 1.340 | 0.632   | 0.620    |
> | BindNet (remove overlap) | 1.396 | 0.625   | 0.616    |
>
> As depicted in the table above, the influence of overlap on the docking task performance is nonnegligible, leading us to report the results in the second row (remove overlap version) in our paper. Conversely, for the binding affinity task, the disparity between the non-overlap and overlap is relatively marginal, and this may be caused by our insufficient pre-training(only 6 epochs due to time limitation vs the orignial 30 epochs) . Consequently, we retained the overlap to maintain alignment with other studies [1], [2] that also utilized Biolip as a pretraining dataset.
>
> [1]CoSP: Co-supervised pretraining of pocket and ligand
>
> [2]General-purpose Pre-trained Model Towards Cross-domain Molecule Learning

---

> ### Author Response · Authors · 2023-11-20
>
> > The authors claim that "a variety of pre-existing encoders for pockets and ligands can be utilized. ", but only Uni-Mol used in experiments.
>
> Thank you for your valuable suggestion. This particular question has been raised by all three reviewers. To answer this question, we have made significant modifications, specifically by replacing both the molecular encoder and pocket encoder. This change aims to demonstrate the adaptability of our proposed framework to a variety of pre-existing encoders.
>
> On the molecular side, we have substituted the **molecular encoder** from UniMol to the pretrained encoder provided by **Frad**[1], which is a molecular pretraining model with a focus on quantum-related tasks. Frad's backbone is built on torchmd-net, which is a gnn and does not output pair representations like 'unimol.' Integrating Frad into our framework is seamless as we can simply use the concatenation of corresponding node-level embeddings as pairwise representations. We conducted pretraining of the interaction module based on Frad for 8 epochs, considering time limitations. We reevaluated the performance of LBA, and additionally, we present results for the setting where the interaction module is unpretrained, as shown in the table below.
>
> On the protein side, we replaced the **pocket encoder** with the **ESM2**[2] encoder and altered the protein input for pocket structure to the protein sequence. Pretraining was also conducted based on the ESM2 encoder. Similarly, we include results for the setting where the interaction module is unpretrained based on the ESM2 encoder, as detailed in the table below.
> | idx | Methods                                | LBA(30%) RMSE | LBA(30%) Pearson | LBA(30%) Spearman | LBA(60%) RMSE | LBA(60%) Pearson | LBA(60%) Spearman |
> | --- | -------------------------------------- | ------------- | ----------------- | ------------------ | ------------- | ----------------- | ------------------ |
> |     | **Replace molecular encoder to Frad**  |               |                   |                    |               |                   |                    |
> | 0   | BindNet(Frad encoder) wo pretrain      | 1.417         | 0.562             | 0.543              | 1.62          | 0.582              | 0.571              |
> | 1   | BindNet(Frad encoder) with pretrain    | 1.386         | 0.612              | 0.597              | 1.341         | 0.761              | 0.762               |
> | 2   | BindNet(UniMol) wo pretrain            | 1.434         | 0.565             | 0.541               | 1.357         | 0.753             | 0.753               |
> | 3   | BindNet(UniMol encoder) with pretrain  | 1.341          | 0.632             | 0.622               | 1.232          | 0.793             | 0.788              |
> |     | **Replace pocket encoder**             |               |                   |                    |               |                   |                    |
> | 4   | BindNet(esm2 encoder) wo pretrain      | 1.561         | 0.444             | 0.445              |               |                   |                    |
> | 5   | BindNet(esm2 encoder) with pretrain    | 1.492         | 0.522             | 0.550               |               |                   |                    |
>
> Upon examining the table above, it is evident that replacing the molecular or pocket encoder with Frad or ESM consistently results in superior performance for the pretrained version of Bindnet compared to the unpretrained version (idx0 vs idx1, idx4 vs idx5). Notably, Bindnet based on Frad continues to achieve SOTA results among other baselines, highlighting the flexibility of our framework.
>
>
> It is essential to draw attention to the ESM2 encoder, which has nearly 3 billion parameters. Due to limitations in computational resources and time constraints, we were compelled to randomly select residues when the sequence number exceeded 500, and the model was trained for only 4 epochs.  Such constraints may have harmful effects on its overall performance. These constraints may have deleterious effects on its overall performance. We firmly believe that alleviating these constraints could significantly enhance its performance.
>
> [1] Fractional Denoising for 3D Molecular Pre-training. ICML 2023
>
> [2] Language models of protein sequences at the scale of evolution enable accurate structure prediction.

---

> ### Author Response · Authors · 2023-11-20
>
> >  Using RDKit to build the primary state is not accurate and sometimes even fail. The authors are encouraged to use more advanced methods for this.
>
> Indeed, relying solely on RDKit to generate the primary state is not always accurate and may even encounter failures. The decision to use the primary state instead of the torsion state is driven by our intent to introduce perturbations to the ligand structure, thereby rendering the ADP task more challenging and meaningful. This approach also serves to enhance the diversity of the pretraining data. The challenge does not lie in the inaccuracy of RDKit, but rather ensuring perturbation in cases where RDKit encounters failure.
>
> Initially, we attempted to leverage RDKit to generate a random stable conformation based on the ligand's chemical information. In instances of failure, we implemented a strategy involving manual introduction of rotations to the dihedral angles by applying Gaussian noise. Subsequently, if the ligand lacked rotatable bonds, we introduced Gaussian noise to the coordinates of the original conformer. To provide a comprehensive understanding of this algorithm, we offer complete pseudocode in Algorithm 1, detailed in the figure accessible via the following link:https://anonymous.4open.science/r/BindNet_rebuttal-88C1/data_perterbation.jpeg
>
> > What is the " N-layer 3D-invariant Transformer" used in the paper? The authors should provide more details about this part.
>
> This question was also raised by reviewer kw5e. We apologize for any confusion, and we understand the need for additional clarification. We will provide a more detailed explanation in our revised paper.
>
> The block in the N-transformer-based interaction module remains consistent with the previously pre-trained molecular encoder. To enhance clarity, we have incorporated a figure within the link https://anonymous.4open.science/r/BindNet_rebuttal-88C1/interaction_module.jpeg, illustrating the module structure to facilitate a better understanding.
>
> As depicted in the figure, we denote the attention weight of the last layer for the pocket encoder as $\mathbf{h}^{(0)} _{PP}$ and the corresponding node-level embedding as $\mathbf{h}^{(0)} _{P}$. Similarly, the molecular attention weight is represented as $\mathbf{h}^{(0)} _{LL}$, and $\mathbf{h}^{(0)} _{L}$ signifies the atom embedding of the molecule.
> The interaction module receives both node-level embeddings and pairwise-level embeddings, the latter serving as the attention bias added to the attention weight. This fusion combines representations from the pocket and ligand. Specifically, we concatenate the pocket and molecular embeddings to form the atom-level input. For the pairwise-level embeddings, denoted as $\mathbf{h}^{(0)} _{PL}$, we initialize it with zero values and subsequently populate it with the values from $\mathbf{h}^{(0)} _{PP}$ and $\mathbf{h}^{(0)} _{LL}$ at the corresponding positions, as follows: $\mathbf{h}^{(0)} _{PL}[:L_p, :L_p]=\mathbf{h}^{(0)} _{PP}$ and $\mathbf{h}^{(0)} _{PL}[-L_m:,-L_m:]=\mathbf{h}^{(0)} _{PP}$. Where the $L_p$ and $L_m$ denotes the atom numbers of pocket and ligand respectively.
> After forwarding through N transformer layers, we obtain the final node-level embeddings $\mathbf{h}^{(N)} _{P}$ and $\mathbf{h}^{(N)} _{L}$, utilized for Mask Ligand Reconstruction, as well as the pairwise embedding $\mathbf{h}^{(N)} _{PL}$, employed for Atomic Pairwise Distance Map Prediction task.
>
> > How to choose the mask ratio in MLR? The 0.8 seems to be a very high value
>
> We selected a higher mask ratio with the intention of relying solely on the pocket to reconstruct the molecular ligand that binds to it. This approach is reminiscent of the pocket-based generation setting[1][2]. However, our Masked Language Reconstruction (MLR) operates at the embedding level, and a larger mask ratio may prove beneficial in enabling molecular or pocket embeddings to encode more interaction-related information. To validate this hypothesis, we set the mask ratio to 0.3 and retrained Bindnet. The results demonstrate that a larger mask ratio indeed improves performance on LBA, as illustrated in the table below.
>
> | Methods                  | RMSE  | Pearson | Spearman |
> | ------------------------ | ----- | ------- | -------- |
> | BindNet (mask ratio=0.3) | 1.370 | 0.619   | 0.608    |
> | BindNet (mask ratio=0.8) | 1.340 | 0.632   | 0.620    |
>
> While the table suggests that a larger mask ratio is more beneficial, it's important to note that determining **the optimal parameter may require further experimentation in a more fine-grained manner**. This process involves time-consuming pretraining, and we plan to conduct additional experiments in the future to identify the best candidate due to time limitations.
>
> [1]:A 3D Generative Model for Structure-Based Drug Design
>
> [2]:Pocket2Mol: Efficient Molecular Sampling Based on 3D Protein Pockets

---

> ### Author Response · Authors · 2023-11-21
>
> Dear reviewer:
> We extend our sincere gratitude for your invaluable suggestions concerning the enhancement of the manuscript. We kindly request your confirmation on whether any outstanding issues remain that we can further attend to in order to align with your expectations and potentially elevate the overall assessment. Your time and feedback are immensely appreciated, and we eagerly anticipate your response.

---

> ### Author Response · Authors · 2023-11-22
>
> Dear reviewer:
>
> If you have any questions, please feel free to contact us at any time. We appreciate your help in improving our work. If our answer has successfully solved your problem, we kindly request that you consider giving us an appropriate increase in score. Your feedback is greatly appreciated and helps us to continue polishing our work.Thank you for your time and consideration. Best regards.

---

> ### Author Response · Authors · 2023-11-23
>
> Thanks for your valuable time spent reviewing our paper. We have thoroughly addressed all of your questions in the following responses and clarified some aspects in the revised version of the paper: https://anonymous.4open.science/api/repo/BindNet_rebuttal-88C1/file/BindNet_revised.pdf. **The primary updates can be found in the appendix and highlighted in blue** for your reference.

---

> ### Author Response · Authors · 2023-11-23
> **Eagerly Awaiting Your Response as Discussion Deadline Approaches**
>
> Dear reviewer:
>
> We extend our heartfelt appreciation for dedicating valuable time to review our paper. Your thoughtful insights and inquiries have been instrumental in refining our work. As the **discussion deadline draws near**, we are eagerly awaiting your response to our recent correspondence. Your insights and feedback are of great importance to us. We appreciate your prompt attention to this matter and look forward to furthering our discussions.
>
> Best regrads

---

> > ### Comment · Reviewer_PEGf · 2023-11-23
> >
> > Thanks for your detailed response. I would like to change my score to 6.

---

### Official Review · Reviewer_kw5e · 2023-10-31

**Soundness:** 2 fair
**Presentation:** 2 fair
**Contribution:** 2 fair
**Rating:** 5
**Confidence:** 4

**Summary:**

This paper introduces a representation learning approach for protein-ligand interactions in a self-supervised learning manner. Inspired by the insufficiencies of previous methods in addressing the interaction module between proteins and ligands, this paper employs a Transformer-based interaction module, designed to emulate the binding process. In addition, the paper suggests two pre-training strategies to optimize the Transformer-based module. The first strategy is an atomic pairwise distance map prediction, grounded in the understanding that various interactions between proteins and ligands correlate with their inter-molecular distances. The second strategy is a mask ligand reconstruction within a feature space, rather than based on atom type, to capture the conditional dependencies between proteins and ligands.

**Strengths:**

- It empirically shows various experiments for protein-ligand binding
- It achieves good performance through simple but effective loss based on the domain knowledge

**Weaknesses:**

* As the major argument of the paper is regarding the capturing of the **interactions** through a Transformer-based interaction module, analysis on the interactions and its consequence should have been provided (e.g., specific types of interactions captured by this new module), which are missing in the paper.
* While it contends that they present a Transformer-based interaction module, the explanation provided is notably insufficient.
* The authors argue that the framework is flexible, allowing for the integration of various pre-existing encoders for proteins and ligands. However, for the distance matrix associated with the atomic pairwise distance map prediction, it appears to necessitate a pair representation in the Uni-Mol. In other words, the architecture seems to be tailored specifically for Uni-Mol.
* If the system is indeed flexible, there should be experimental evidence presented with other encoders.

**Questions:**

* BindNet was pre-trained using BioLip. However, were the other baseline models also pre-trained on BioLip? Notably, the pre-trained model weights for Uni-Mol, available on their GitHub, seem to have utilized other datasets. If Uni-Mol, initially pre-trained on other datasets, was subsequently pre-trained using BioLip, this could create an inequity with the other baselines due to the discrepancy in the number of datasets used for pre-training.
* Did the other baseline models use the primary state for ligands in experiments?

---

> ### Author Response · Authors · 2023-11-20
> **Official Comment by Authors**
>
> Thank you sincerely for dedicating your time to review our paper. We highly value your expert feedback and insightful concerns. Your suggestions have been thoroughly considered, and we have provided detailed explanations where necessary.
>
> >Analysis on the interactions and its consequence should have been provided (e.g., specific types of interactions captured by this new module), which are missing in the paper.
>
> Thank you for your suggestions. We have selected four pocket-ligand complexes as examples to demonstrate the information captured by our interaction module in the following links(https://anonymous.4open.science/r/BindNet_rebuttal-88C1/sch1.jpeg; https://anonymous.4open.science/r/BindNet_rebuttal-88C1/sch2.jpeg; https://anonymous.4open.science/r/BindNet_rebuttal-88C1/sch3.jpeg; https://anonymous.4open.science/r/BindNet_rebuttal-88C1/sch4.jpeg).
>
> The left part of figure illustrates interactions computed using **Ligand Interaction Diagram Panel of Schrödinger**[Schrödinger, LLC, NY, USA], depicting three common interactions: **hydrogen interaction** represented by the magenta line, **π-π interaction** denoted by the green line, and **salt bridge** visualized as a gradient color line transitioning from red to blue.
>
> The right part of figure represents the attention heatmap between pocket and ligand from BindNet. The original attention in BindNet is atom-atom level, which is too long to present here due to the significant number of atoms in the pocket, so we aggregate the attention of same residue in pocket to form a atom-residue level attention heatmap for a clearer visual representation.
> From these examples, we can easily find that **BindNet pays more attention to the residue which can form non-covalent interactions with ligand**, sometimes even the specific ligand atom - residue pair. This implys BindNet might have the ability to capture the interaction force information between the pocket and ligand, which is crucial for the task such as ligand binding affinity prediction and virtural screening.
>
>
>
> > While it contends that they present a Transformer-based interaction module, the explanation provided is notably insufficient.
>
> Thank you for your reminder. In the revised version, we will include a more detailed explanation. The block in the N-transformer-based interaction module remains consistent with the previously pre-trained molecular encoder. To enhance clarity, we have incorporated a figure within the link https://anonymous.4open.science/r/BindNet_rebuttal-88C1/interaction_module.jpeg, illustrating the module structure to facilitate a better understanding.
>
> As depicted in the figure, we denote the attention weight of the last layer for the pocket encoder as $\mathbf{h}^{(0)} _{PP}$ and the corresponding node-level embedding as $\mathbf{h}^{(0)} _{P}$. Similarly, the molecular attention weight is represented as $\mathbf{h}^{(0)} _{LL}$, and $\mathbf{h}^{(0)} _{L}$ signifies the atom embedding of the molecule.
> The interaction module receives both node-level embeddings and pairwise-level embeddings, the latter serving as the attention bias added to the attention weight. This fusion combines representations from the pocket and ligand. Specifically, we concatenate the pocket and molecular embeddings to form the atom-level input. For the pairwise-level embeddings, denoted as $\mathbf{h}^{(0)} _{PL}$, we initialize it with zero values and subsequently populate it with the values from $\mathbf{h}^{(0)} _{PP}$ and $\mathbf{h}^{(0)} _{LL}$ at the corresponding positions, as follows: $\mathbf{h}^{(0)} _{PL}[:L_p, :L_p]=\mathbf{h}^{(0)} _{PP}$ and $\mathbf{h}^{(0)} _{PL}[-L_m:,-L_m:]=\mathbf{h}^{(0)} _{PP}$. Where the $L_p$ and $L_m$ denotes the atom numbers of pocket and ligand respectively.
> After forwarding through N transformer layers, we obtain the final node-level embeddings $\mathbf{h}^{(N)} _{P}$ and $\mathbf{h}^{(N)} _{L}$, utilized for Mask Ligand Reconstruction, as well as the pairwise embedding $\mathbf{h}^{(N)} _{PL}$, employed for Atomic Pairwise Distance Map Prediction task.

---

> ### Author Response · Authors · 2023-11-20
> **Official Comment by Authors**
>
> > If the system is indeed flexible, there should be experimental evidence presented with other encoders.
>
> Thank you for your valuable suggestion. This particular question has been raised by all three reviewers. To answer this question, we have made significant modifications, specifically by replacing both the molecular encoder and pocket encoder. This change aims to demonstrate the adaptability of our proposed framework to a variety of pre-existing encoders.
>
> On the molecular side, we have substituted the **molecular encoder** from UniMol to the pretrained encoder provided by **Frad**[1], which is a molecular pretraining model with a focus on quantum-related tasks. Frad's backbone is built on torchmd-net, which is a gnn and does not output pair representations like 'unimol.' Integrating Frad into our framework is seamless as we can simply use the concatenation of corresponding node-level embeddings as pairwise representations. We conducted pretraining of the interaction module based on Frad for 8 epochs, considering time limitations. We reevaluated the performance of LBA, and additionally, we present results for the setting where the interaction module is unpretrained, as shown in the table below.
>
> On the protein side, we replaced the **pocket encoder** with the **ESM2**[2] encoder and altered the protein input for pocket structure to the protein sequence. Pretraining was also conducted based on the ESM2 encoder. Similarly, we include results for the setting where the interaction module is unpretrained based on the ESM2 encoder, as detailed in the table below.
> | idx | Methods                                | LBA(30%) RMSE | LBA(30%) Pearson | LBA(30%) Spearman | LBA(60%) RMSE | LBA(60%) Pearson | LBA(60%) Spearman |
> | --- | -------------------------------------- | ------------- | ----------------- | ------------------ | ------------- | ----------------- | ------------------ |
> |     | **Replace molecular encoder to Frad**  |               |                   |                    |               |                   |                    |
> | 0   | BindNet(Frad encoder) wo pretrain      | 1.417         | 0.562             | 0.543              | 1.62          | 0.582              | 0.571              |
> | 1   | BindNet(Frad encoder) with pretrain    | 1.386         | 0.612              | 0.597              | 1.341         | 0.761              | 0.762               |
> | 2   | BindNet(UniMol) wo pretrain            | 1.434         | 0.565             | 0.541               | 1.357         | 0.753             | 0.753               |
> | 3   | BindNet(UniMol encoder) with pretrain  | 1.341          | 0.632             | 0.622               | 1.232          | 0.793             | 0.788              |
> |     | **Replace pocket encoder**             |               |                   |                    |               |                   |                    |
> | 4   | BindNet(esm2 encoder) wo pretrain      | 1.561         | 0.444             | 0.445              |               |                   |                    |
> | 5   | BindNet(esm2 encoder) with pretrain    | 1.492         | 0.522             | 0.550               |               |                   |                    |
>
> Upon examining the table above, it is evident that replacing the molecular or pocket encoder with Frad or ESM consistently results in superior performance for the pretrained version of Bindnet compared to the unpretrained version (idx0 vs idx1, idx4 vs idx5). Notably, Bindnet based on Frad continues to achieve SOTA results among other baselines, highlighting the flexibility of our framework.
>
>
> It is essential to draw attention to the ESM2 encoder, which has nearly 3 billion parameters. Due to limitations in computational resources and time constraints, we were compelled to randomly select residues when the sequence number exceeded 500, and the model was trained for only 4 epochs.  Such constraints may have harmful effects on its overall performance. These constraints may have deleterious effects on its overall performance. We firmly believe that alleviating these constraints could significantly enhance its performance.
>
> [1] Fractional Denoising for 3D Molecular Pre-training. ICML 2023
>
> [2] Language models of protein sequences at the scale of evolution enable accurate structure prediction.
>
> > BindNet was pre-trained using BioLip. However, were the other baseline models also pre-trained on BioLip?
>
> Another pretrained method, CoSP [1], also utilizes BioLip data as its pretraining data (a comparison with CoSP on DUDE can be found in Table 3). However, CoSP excludes complex structure information and relies solely on pairwise information as a supervisory signal.
>
> [1] CoSP: Co-supervised pretraining of pocket and ligand

---

> ### Author Response · Authors · 2023-11-20
>
> > If Uni-Mol, initially pre-trained on other datasets, was subsequently pre-trained using BioLip, this could create an inequity with the other baselines due to the discrepancy in the number of datasets used for pre-training.
>
> Given that our current experiment is based on Uni-Mol pretrained encoders, we continue to pretrain the molecular and pocket encoders of unimol using the BioLip data with the original strategy—coordinate denoising and atom type masking. **This approach aims to eliminate discrepancies in the number of datasets used for pretraining**. Specifically, both the molecular and pocket encoders are pretrained for 16 and 10 epochs, respectively, until their loss converges. We evaluate the model on the LBA 30% dataset and report the results in the second row of the table below.
>
> | Methods                     | RMSE  | Pearson | Spearman |
> | ---------------------------- | ----- | ------- | -------- |
> | Uni-Mol (origin data)        | 1.434 | 0.565   | 0.54     |
> | Uni-Mol (add biolip data)    | 1.435 | 0.565   | 0.548    |
> | BindNet                      | 1.34  | 0.632   | 0.62     |
>
> As illustrated in the table above, it can be inferred that the addition of BioLip data has no discernible impact on performance, as indicated by the comparable results in the second row versus the first row. Notably, **the scale of the BioLip data (approximately 0.4 million) is trivial when compared to the scale of the original unimol pretraining data (209 million molecular conformations and 3 million pockets)**. **The performance gap may be attributed more to differences in pretraining methods than to the increase in pretraining data**.
>
> Other pretraining baselines employ different types of pretraining data compared to ours, as different datasets are suitable for distinct pretraining strategies. For instance, SMT-DTA utilizes Masked Language Modeling (MLM) on sequences of molecules and proteins, drawing from 10 million unpaired molecular and protein sequences from PUBCHEM and Pfram. Meanwhile, GeoSLL employs distance denoising as its pretraining task, relying on Density Functional Theory (DFT)-calculated equilibrium conformations from PubChemQC. The selection of different types of pretraining data is dictated by the design of self-supervised methods, making it challenging to align them with the same number of datasets for direct comparison.
>
>
>
> > Did the other baseline models use the primary state for ligands in experiments?
>
> For **binding affinity tasks** (LBA and LEP, Section 4.1), where complex data is provided as input, all baseline models utilize the torsion state (molecular structure in the binding complex) of ligands. In contrast, for **virtual screening tasks** (DUD-E, DUD-AD, Section 4.3) and the **docking task** (Section 4.2), the complex structure is either unavailable or can only be used as a label for prediction. In these cases, all methods use the primary state of ligands as input.
>
> Notably, for the binding affinity tasks, unlike other structure-based methods, BindNet can discern the dependency on torsion state. Whether considering the torsion state or primary state, neither influences the state-of-the-art (SOTA) performance of BindNet, as confirmed by ablation studies in Section 5.2.

---

> ### Author Response · Authors · 2023-11-21
>
> Thank you for your invaluable suggestions regarding the manuscript's improvement. We have thoroughly reviewed your feedback and  addressed your questions in detail and integrated more extensive and unbiased experimental evidence. We would appreciate it if you could confirm whether there are any remaining issues that we can further address to meet your expectations and potentially enhance the overall assessment. Your time and feedback are highly valued, and we eagerly await your response.

---

> > ### Comment · Reviewer_kw5e · 2023-11-22
> >
> > Thank you for addressing my concerns with your response. I raised my score accordingly. Please update the results in the paper regarding the discussed contents.
> > However, regarding the qualitative analysis for pocket-ligand complexes, it is difficult to demonstrate whether the effectiveness is from BindNet or the pre-trained model because only BindNet's results are provided. Because the pre-trained model could capture some relations between pocket and ligand, if the authors claim that "BindNet pays **more** attention to the residue ~", the comparison between the results of both before and after applying BindNet should have been provided. Based on the comparison, I would like to raise my score more. Best regards,

---

> ### Author Response · Authors · 2023-11-22
> **Comparsion between the attention results of both before and after applying BindNet**
>
> Thank you for your thoughtful review and valuable suggestions. Your insights have significantly contributed to improving our paper. We will incorporate the discussed contents in our revised paper.
>
> We have introduced a comparison with the settings before applying BindNet pretraining strategy(denoted as "Without BindNet"), utilizing the **same four protein-ligand complexes presented earlier**. The figures associated with this comparison can be accessed through the following links:
> https://anonymous.4open.science/r/BindNet_rebuttal-88C1/sch1c.jpeg
> https://anonymous.4open.science/r/BindNet_rebuttal-88C1/sch2c.jpeg
> https://anonymous.4open.science/r/BindNet_rebuttal-88C1/sch3c.jpeg
> https://anonymous.4open.science/r/BindNet_rebuttal-88C1/sch4c.jpeg
>
> In each figure, the left part continues to depict common interactions computed by the Ligand Interaction Diagram Panel of Schrödinger, as demonstrated previously. The right part illustrates two types of attention heatmaps generated by models with and without applying BindNet.
> Upon comparing the two, it is evident that **the attention map generated by model without applying BindNet appears random and lacks discernible correlations with the interactions presented in the right part**. This observation is reasonable given that the pretrained model (pocket encoder and molecular encoder) is trained on individual molecules or pockets, making it unable to capture binding information without applying the BindNet pretraining strategy based on the complex data, specifically designed to learn the fine-grained interactions between the pocket and ligand.

---

> ### Author Response · Authors · 2023-11-23
>
> Thanks for your valuable time spent reviewing our paper. We have thoroughly addressed all of your questions in the following responses and clarified some aspects in the revised version of the paper: https://anonymous.4open.science/api/repo/BindNet_rebuttal-88C1/file/BindNet_revised.pdf. **The primary updates can be found in the appendix and highlighted in blue** for your reference.

---

> ### Author Response · Authors · 2023-11-23
> **Eagerly Awaiting Your Response as Discussion Deadline Approaches**
>
> Thank you for your valuable suggestions on how to improve the manuscript. We have carefully considered your feedback and made the necessary revisions to the paper. In particular, we have provided detailed answers to your questions and incorporated more comprehensive and impartial experimental evidence.  **As the discussion deadline draws near**, we would like to confirm if there are any remaining issues that we can address to ensure your satisfaction with the revisions and potentially increase the score. We appreciate your time and feedback, and look forward to your response.
>
> Thank you for your time and consideration!
>
> Best regards

---

> ### Author Response · Authors · 2023-11-23
> **Eagerly Awaiting Your Response as Discussion Deadline Approaches**
>
> Dear Reviewer,
>
> **In response to your valuable suggestions, we have incorporated a comparison with the settings before applying BindNet pretraining strategy illustrate BindNet enables the model to learn interaction patterns that accurately reflect real binding pocket-ligand forces**. Furthermore, we have integrated the relevant discussion content into a revised version of the paper.
>
> **As the deadline for discussions approaches, we sincerely hope that our responses have effectively addressed any concerns or questions you may have had. We kindly request your consideration for an appropriate increase in the score if you find our amendments satisfactory**. Your constructive feedback is immensely valuable to us and contributes significantly to the refinement of our work.
>
> Thank you for your time, attention, and thoughtful consideration.
>
> Best Regards,

---

### Meta-Review · Area_Chair_XV6n · 2023-12-07

**Metareview:**

The paper proposes a novel self-supervised approach for representation learning of protein-ligand binding, along with two pre-training tasks, and demonstrate the proposed framework on several down-stream tasks.

The authors have done a great job at addressing the reviewers concerns. The extra experiments and materials significantly improve the significance of the manuscript.

**Justification For Why Not Higher Score:**

Although the approach is novel, it is not groundbreaking as the overall framework of self-supervised learning and the proposal of pre-training objectives is now well established in the context of protein representation learning.

**Justification For Why Not Lower Score:**

The paper studies an interesting problem. The approach is convincing and experimental study quite rich.

---

### Decision · Program_Chairs · 2024-01-16

Accept (poster)